# Simple Yet Effective: Structure Guided Pre-trained Transformer for Multi-modal Knowledge Graph Reasoning

## ABSTRACT

Various information in different modalities in an intuitive way in multi-modal knowledge graphs (MKGs), which are utilized in different downstream tasks, like recommendation. However, most MKGs are still far from complete, which motivates the flourishing of MKG reasoning models. Recently, with the development of general artificial intelligence, pre-trained transformers have drawn increasing attention, especially in multi-modal scenarios. However, the research of multi-modal pre-trained transformers (MPT) for knowledge graph reasoning (KGR) is still at an early stage. As the biggest difference between MKG and other multi-modal data, the rich structural information underlying the MKG is still not fully utilized in previous MPT. Most of them only use the graph structure as a retrieval map for matching images and texts connected with the same entity, which hinders their reasoning performances. To this end, the graph **S**tructure **G**uided **M**ulti-modal **P**re-trained **T**ransformer is proposed for knowledge graph reasoning (SGMPT). Specifically, the graph structure encoder is adopted for structural feature encoding. Then, a structure-guided fusion module with two simple yet effective strategies, *i.e.,* weighted summation and alignment constraint, is designed to inject the structural information into both the textual and visual features. To the best of our knowledge, SGMPT is the first MPT for multi-modal KGR, which mines structural information underlying MKGs. Extensive experiments on FB15k-237-IMG and WN18-IMG, demonstrate that our SGMPT outperforms existing state-of-the-art models, and proves the effectiveness of the designed strategies.

## CCS CONCEPTS

• **Information systems** → **Multimedia information systems**; • **Computing methodologies** → **Knowledge representation and reasoning**.

## KEYWORDS

Knowledge Graph Reasoning, Multimodal Information Fusion, Pre-trained Transformer Model

### ACM Reference Format:
Anonymous Authors. 2024. Simple Yet Effective: Structure Guided Pre-trained Transformer for Multi-modal Knowledge Graph Reasoning. In *Proceedings of Make sure to enter the correct conference title from your rights confirmation emai (ACM MM '24).* ACM, New York, NY, USA, 9 pages. https://doi.org/XXXXXXX.XXXXXXX

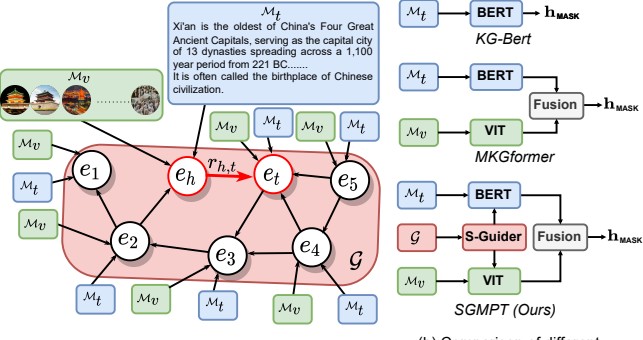

(a) Illustration of multi-modal KG

(b) Comparison of different pretrained transformer KGR models

**Figure 1: Illustration of multi-modal knowledge graph and existing pre-trained transformer KGR models. $\mathcal{M}_t$, $\mathcal{M}_v$, and $\mathcal{G}$ represent the textual information, visual information, and graph structural information, respectively. Besides, BERT [12] and ViT [14] are two commonly used transformers for encoding $\mathcal{M}_t$ and $\mathcal{M}_v$, and structure-guided fusion module (S-Guider) is a novel module designed in our SGMPT to leverage $\mathcal{G}$. Note that $e$ and $r$ denote entity and relation separately.**

## 1 INTRODUCTION

Multi-modal knowledge graphs (MKGs), which intuitively organize information in various modalities, can benefit many practical downstream tasks, such as recommendation systems [20, 52], information retrieval [13, 49], and visual question answering [21]. Specifically, compared to traditional KGs, extra multi-modal data are linked with entities in MKGs to provide more meaningful information, such as visual and textual attributes, which makes them closer to the real world. However, existing multi-modal KGs may suffer from even more severe incompleteness issues due to the insufficient accumulation of multi-modal corpus. It compromises their usability and impairs their effectiveness. To address the limitations, more and more recent attempts [7, 44, 45, 47, 54, 55] for knowledge graph reasoning (KGR) are taken in multi-modal scenarios, and our work also lays in this scope.

The incompleteness issues, as a common nature among different types of KGs, have been widely studied these years [28]. According to previous attempts for KGR, we can easily draw a conclusion that the performance of the KGR model highly depends on whether the information in KGs is fully leveraged or not. For example, the conventional KGR models [9, 38, 42, 51] focus on mining the structural information in static KGs. However, an additional effective mechanism for temporal information is important for these models to achieve better reasoning in temporal KGs. It is similar to multi-modal knowledge graph reasoning (MKGR), which requires specific

mechanisms to mine the multi-modal information since previous unimodal KGR models can also not scale well to multi-modal scenarios. Following this trace, different mechanisms are designed based on the unimodal KGR models to fuse the heterogeneous multi-modal features of each entity into a unified representation, which builds up various multi-modal KGR models. For example, IKRL [46] and MTRL [31] are both developed based on TransE [4] but with different fusion strategies. The former model adopts the attention mechanism to integrate the visual information, while the fusion module in the latter one offers three different fusion options. However, with the development of general artificial intelligence (AGI), the pre-trained transformer architecture has drawn increasing attention as a more general and powerful paradigm, especially for multi-modal scenarios. Inspired by their success in other fields, lots of pre-trained transformer models for KGR have come out these years, such as KG-BERT [50], KGformer [29], and etc. However, the research on developing an effective multi-modal pre-trained transformer (MPT) for KGR is still at an early stage, which leaves us a huge gap to explore.

Among the MPT models for KGR, MKGformer [7] is the most representative model with the best reasoning capacity. Although achieving promising performance for MKGR, it still fails to fully leverage the structural information underlying the knowledge graph, which hinders their reasoning capacity. Unlike other multi-modal data, MKG usually contains three types of information, *i.e.,* textual information $\mathcal{M}_t$ (*e.g.,* text description), visual information $\mathcal{M}_v$ (*e.g.,* images), and graph structural information $\mathcal{G}$ as shown in Fig. 1 (a). MPT models for other multi-modal scenarios only take consideration of the first two modalities, which is also how MKGformer does for MKGR task as shown in Fig. 1 (b). More specifically, the graph structure of MKG is only utilized as a retrieval map for matching images and text descriptions corresponding to the same entity. In this manner, the rich structural information underlying the graph structure $\mathcal{G}$ is ignored, such as the relational information between different entities and the topological information within the graph structure. This structural information will definitely benefit the expressive ability of the models, which has been proven in those multi-modal non-transformer KGR models [44, 54]. As it is currently ignored by MPT models for MKGR, all we need is to design an effective fusion mechanism to mine such structural information, which will endow the existing MPT models with greater capacity for better reasoning performance.

Following this idea, we propose a novel graph Structure Guided Multi-modal Pre-trained Transformer model for knowledge graph reasoning, termed SGMPT, by designing a plug-and-play mechanism to leverage the structural information omitted by previous MPT models as shown in Fig. 2. More specifically, the graph structure encoder is adopted for structure feature encoding. Then, a structure-guided fusion module with two different strategies, *i.e.,* weighted summation and alignment constraint, is first designed to fuse the structure information in both the textual and visual features. Concretely, (1) weighted summation directly adds the generated structural embedding with the textual and visual embeddings in the segment for the entity. Besides, (2) the alignment constraint adopts the alignment loss, *i.e.,* MSE loss [15, 39], to guide the learning procedure by refining original textual and visual embeddings according to structural information. Moreover, the above strategies

can be adopted both individually and composedly. To the best of our knowledge, SGMPT is the first multi-modal pre-trained transformer for KGR, which tries to mine the structural information underlying the knowledge graph. In addition, extensive experiments are carried out on two typical benchmark datasets to demonstrate the promising capacity of SGMPT from four aspects, *i.e.,* superiority, effectiveness, efficiency, and sensitivity. The main contributions are summarized below:

- We propose a novel and simple graph structure guided multi-modal pre-trained transformer model for knowledge graph reasoning, termed SGMPT, by effectively making use of the knowledge graph structural information.
- We adopt the structural encoder and design a plug-and-play mechanism, *i.e.,* structure-guided fusion module, is proposed to complement the omitted graph structural information for MPT models for KGR. Specifically, the structure-guided fusion module contains two different strategies, *i.e.,* weighted summation and alignment constraint, which can be adopted both individually and composedly.
- Extensive experiments on FB15k-237-IMG and WN18-IMG datasets demonstrate the capacity of SGMPT from four aspects, *i.e.,* superiority, effectiveness, efficiency, and sensitivity, and also proves the effectiveness of the structural information. The codes will be open-sourced after the review procedure.

## 2 RELATED WORK

### 2.1 Multi-modal Knowledge Graph Reasoning

Multi-modal knowledge graph reasoning (MKGR) aims to infer the potential missing facts in multi-modal knowledge graphs, which can be roughly divided into two types, *i.e.,* non-transformer models and transformer models, according to the model architectures.

*2.1.1 Non-Transformer Multi-modal KGR Models.* Most of the MKGR models are developed based on non-transformer architectures. Different mechanisms are designed to encode the extra modal information by extending the original unimodal KGR models, such as TransE [4]. For example, IKRL [46] first adopts the attention-based mechanism to integrate the visual information and the original structural information generated by the translation-based KGR models. MTRL [31] offers three different strategies, *i.e.,* simple summation, DeViSE [16], and Imagined [10] to integrate multi-modal information. In addition, TransAE [44] utilized an auto-encoder to use them. Moreover, MoSE [54] exploits three ensemble inference techniques to combine the modality-split predictions by assessing modality importance. Recently, RSME [43] designed a forget gate with an MRP metric to select valuable images for multi-modal KGR, which tries to avoid the influence caused by the noise from irrelevant images corresponding to entities. However, our model does not belong to this type.

*2.1.2 Transformer Multi-modal KGR Models.* The transformer models originated form natural language processing [5, 12, 33], and quickly shifted the paradigm of image processing [3, 14] from fully supervised learning to pre-training and fine-tuning. Due to their promising capabilities in multi-modal scenarios, various general multi-modal pre-trained transformer (MPT) models [8, 18, 24, 26,

30, 35, 37] have been proposed these years. However, the target optimization objects of the above multi-modal pre-trained models are less relevant to knowledge graph reasoning (KGR) tasks. Due to the variance between the multi-modal knowledge graph (MKG) and other multi-modal data, directly applying the above general MPT models to multi-modal knowledge graph reasoning (MKGR) may not lead to good reasoning [7].

Meanwhile, pre-trained transformer models [6, 7, 19, 23, 25, 29, 36] for KGR are also springing up, such as KG-BERT [50], which is the first pre-trained contextual language model for the KGR task, etc. However, the research on developing an effective multi-modal pre-trained transformer (MPT) for KGR is still at an early stage. Among them, MKGformer [7] is the most representative attempt with promising reasoning capacity, which leaves us huge space to explore better MPT models for KGR. But MKGformer [7] also ignores the key difference of the characteristic between MKG and other multi-modal data. Specifically, unlike other multi-modal data, MKG usually contains three types of information, *i.e.,* textual information $\mathcal{M}_t$ (*e.g.,* text description), visual information $\mathcal{M}_v$ (*e.g.,* images), and graph structural information $\mathcal{G}$ as shown in Fig. 1 (a). MPT models for other multi-modal scenarios only take consideration of the first two modalities, which is also how MKGformer does for MKGR task as shown in Fig. 1 (b). In other words, the graph structure of MKG is only utilized as a retrieval map for matching images and text descriptions corresponding to the same entity. In this manner, the rich structural information underlying the knowledge graph $\mathcal{G}$ is ignored, such as the relational information between different entities and the topological information within the graph structure. This structural information will definitely benefit the expressive ability of the models, which has been proven in those multi-modal non-transformer KGR models [44, 54]. As it is currently ignored by MPT models for MKGR, all we need is to design an effective fusion mechanism to mine such structural information for multi-modal transformer KGR models. To this end, our work endows the existing MPT models with greater capacity for those omitted structural information to achieve better reasoning.

## 2.2 Multi-modal Fusion Strategy

Information fusion aims to integrate information from different modalities to contribute to the specific downstream tasks [2], which is usually treated as one of the important step for multi-modal and multisource tasks. There are generally two ways to fuse the features, *i.e.,* (1) combining every single modal representation in its own feature space, such as summation, average pooling [1, 40, 56], and (2) learning the unified representations by projecting different modal representations into the same latent space based on a well-designed objective function [31, 32, 46]. Inspired by them, our SGMPT is the first MPT model to fuse the structural information with the original textual and visual information by introducing a plug-and-play structure-guided fusion module. Two strategies are designed in the structure-guided fusion module, including weighted summation and alignment constraint. Concretely, weighted summation belongs to the first type, while the alignment constraint belongs to the second type. Both the above two types of strategies can be adopted individually and composedly. More details about the fusion strategies are illustrated in Section III.C.(2).

## 3 METHOD

In this section, we will introduce the details of the proposed graph structure guided multi-modal pre-trained transformer model, termed SGMPT, from three aspects, *i.e.,* preliminary, multi-modal pre-trained transformer backbone, and guidance of structure information. The overall framework of our SGMPT is shown in Fig. 2, and the pseudo-code of our SGMPT is shown in Algorithm 1.

### 3.1 Preliminary

In this section, we introduce prior knowledge for the multi-modal knowledge graphs and the reasoning task over them.

The multi-modal knowledge graph is defined as a directed graph $MKG = (\mathcal{E}, \mathcal{R}, \mathcal{G}, \mathcal{A}_\mathcal{M})$, where $\mathcal{E}$ and $\mathcal{R}$ represent the entity set and the relation set respectively, and $\mathcal{G} = \{(e_h, r_{h,t}, e_t) \mid e_h, e_t \in \mathcal{E}, r_{h,t} \in \mathcal{R}\}$ is the set of fact triplets. $\mathcal{A}_\mathcal{M}$ represents the set of multi-modal attributes corresponding to each entity, which contains two types of modalities, *i.e.,* text descriptions ($\mathcal{M}_t$) and image descriptions ($\mathcal{M}_v$).

As for the reasoning task over multi-modal KGs, it is defined in this paper as follows. Given the missing facts $(e_h, r_{h,t}, ?)$, the main goal of knowledge graph reasoning (MKGR) is to infer the entity $e_t$ based on the *MKG*. Similar to previous multi-modal pre-trained transformer (MPT) KGR models [7], the MKGR tasks in this paper can be divided into two sub-tasks, including: (1) **Pre-training**: image-text incorporated entity representation learning, and (2) **Fine-tuning**: relation reasoning over multi-modal entity representations. Both of the sub-tasks are trained on multi-modal knowledge graph datasets. As shown in Eq. (1) and Eq. (2), the [CLS] and [SEP] are two separation symbols, and [MASK] is the prediction symbol. Notably, the multi-modal attributes, *i.e.,* visual and structural features, are encoded and integrated into the textual feature, and both pre-training and fine-tuning tasks are still reformulated as masked language modeling (MLM) tasks based on the textual features.

**Pre-training**. The pre-training procedure aims to match the multi-modal attributes with the corresponding masked entity $e_i$.

$$T(e_i) = [\text{CLS}] \; A_i^t \; \text{is the description of} \; [\text{MASK}] [\text{SEP}] \quad (1)$$

**Fine-tuning**. For reasoning task $T(e_h, r_{h,t}, ?)$, the main goal of the fine-tuning procedure is to predict the masked target entity $e_t$.

$$T(e_h, r_{h,t}, ?) = [\text{CLS}] \; e_h \, A_h^t \; [\text{SEP}] \; r_{h,t} \; [\text{SEP}] [\text{MASK}] [\text{SEP}] \quad (2)$$

### 3.2 MPT Backbone

The MKGformer [7], which is the most representative MPT model for KGR, is selected as our MPT backbone. It consists of three different encoders, *i.e.,* text encoder, vision encoder, and multi-modal information encoder. Specifically, the number of layers in the text encoder, vision encoder, and multi-modal information encoder are $L_t$, $L_v$, and $L_m$, respectively, where $L_{\text{BERT}} = L_t + L_m$ and $L_{\text{ViT}} = L_v + L_m$. We briefly introduce the important components of the backbone model as shown below. (See [7] for more details).

**Text Encoder**. Text encoder $f_t(\cdot)$ is composed of the first $L_t$ layers of BERT [12], which aims to capture basic syntactic and lexical information. It takes tokens in the text descriptions $\mathcal{M}_t$ as input,

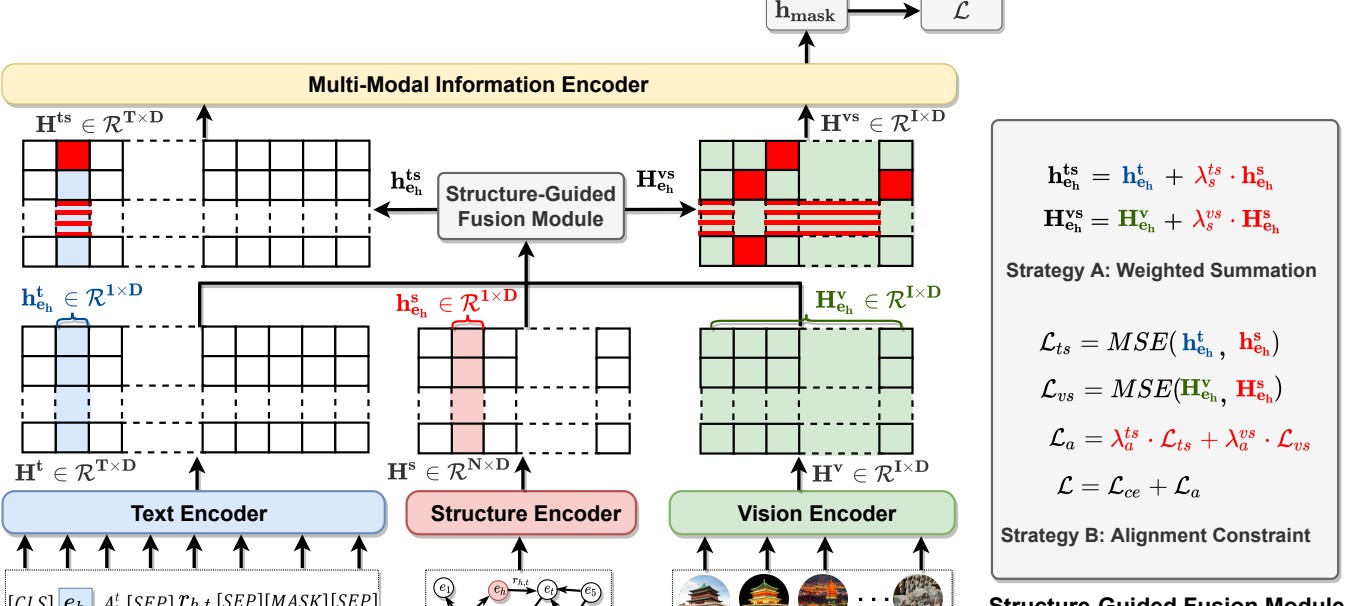

**Figure 2: The framework of the proposed graph Structure Guided Multi-modal Pre-trained Transformer model for knowledge graph reasoning, termed SGMPT. The structure encoder and structure-guided fusion module are proposed to complement the omitted graph structural information for MPT models for knowledge graph reasoning. Precisely, the structure-guided fusion module is a plug-and-play mechanism, which contains two different strategies, *i.e.,* weighted summation and alignment constraint. The blue, green, and red colors represent the textual, visual, and structural features, respectively. Note that more details of the fusion strategies can be found in Fig. 3 and Tab. 1 presents the descriptions of notations.**

and outputs the textual feature $\mathbf{H}^t$.

$$\mathbf{H}^t = f_t(\mathcal{M}_t) \tag{3}$$

**Vision Encoder**. Vision encoder $f_v(\cdot)$ is composed of the first $L_v$ layers of ViT [14], which aims to capture basic visual features from the patched images. It takes images $\mathcal{M}_v$ as input, and outputs the visual feature $\mathbf{H}^v$.

$$\mathbf{H}^v = f_v(\mathcal{M}_v) \tag{4}$$

**Multi-modal Information Encoder**. Following [7], multi-modal information encoder $f_m(\cdot)$ aims to model multi-modal features of the entity across the last $L_m$ layers of ViT and BERT with multi-level fusion. It takes learned representations from previous encoders as input, and outputs the multi-modal representations for inference.

## 3.3 Guidance of Structure Information

To guide the learning procedure for the multi-modal pre-trained transformer (MPT) model for KGR with the structural information, two novel modules are proposed, *i.e.,* structure encoder and structure-guided fusion module. The structure encoder aims to encode the structural information into the feature vector, and the structure-guided fusion module tends to fuse the omitted structural information into the existing MPT models for KGR.

**Table 1: The Summary of important notations**

| Notations | Descriptions |
|---|---|
| $MKG = (\mathcal{E}, \mathcal{R}, \mathcal{G}, \mathcal{A}_\mathcal{M})$ | multi-modal knowledge graph |
| $\mathcal{E}$ | entity set |
| $\mathcal{R}$ | relation set |
| $\mathcal{G}$ | graph with fact triplets (edges) |
| $\mathcal{A}_\mathcal{M}$ | multi-modal attribute set |
| $\mathcal{M}_t, \mathcal{M}_v$ | textual and visual modality |
| $A_h^t = \{w_1, w_2, \cdots, w_n\}$ | textual attribute, $w_i$ represents the $i^{th}$ word |
| $A_h^v = \{I_1, I_2, \cdots, I_m\}$ | visual attribute, $I_i$ represents the $i^{th}$ image |
| $\mathbf{H}, \mathbf{h}$ | feature matrix, feature vector |
| $\mathbf{H}^t \in \mathcal{R}^{T \times D}$ | textual feature matrix, $T$ is the number of tokens |
| $\mathbf{H}^v \in \mathcal{R}^{I \times D}$ | visual feature matrix, $I$ is the number of images |
| $\mathbf{H}^s \in \mathcal{R}^{N \times D}$ | structural feature matrix, $N$ is the number of entities |
| $\mathbf{H}^{ts}, \mathbf{H}^{vs}$ | text-structure and vision-structure feature matrix |
| $\mathbf{h}_{mask}$ | representation of [MASK] |
| $\mathcal{L}$ | total loss |
| $\mathcal{L}_{ts}, \mathcal{L}_{vs}$ | text-structure and vision-structure alignment loss |
| $\mathcal{L}_a, \mathcal{L}_{ce}$ | alignment loss, cross-entropy loss |
| $\lambda_s^{ts}, \lambda_s^{vs}$ | hyperparameter for weighted summation |
| $\lambda_a^{ts}, \lambda_a^{vs}$ | hyperparameter for alignment constraint |

*3.3.1 Structure Encoder.* The structure encoder takes the static KG $\mathcal{G}$ as input without multi-modal attributes and outputs the structural representation $\mathbf{H}^s \in \mathcal{R}^{N \times D}$ for fusion, where $N$ represents the number of the entities in KG. Different knowledge graph embedding models contribute to the candidate structure encoder $g(\cdot)$, such as HAKE [53], ComPGCN [42], Nodepiece [17], etc.

$$\mathbf{H}^s = g(\mathcal{G}) \tag{5}$$

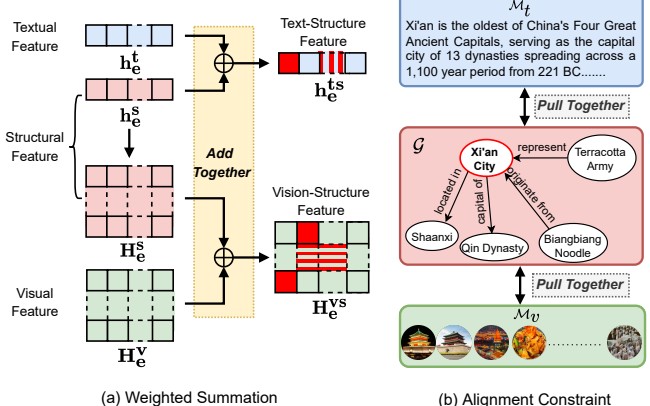

(a) Weighted Summation    (b) Alignment Constraint

**Figure 3: The illustration of the designed strategies in the structure-guided fusion module with an example of the entity 'Xi'an City' (See Tab. 1 for notations).**

This paper selects one of the state-of-the-art models, HAKE [53], as the structure encoder to generate the structural features for most of the experiments. In addition, we also present the influence of different structure encoders in Sec. 4.3.2.

*3.3.2 Structure-Guided Fusion Module.* The structure-guided fusion module is designed to fuse the structural information. Two simple yet effective strategies are designed in the structure guide, *i.e.,* weighted summation, and alignment constraint, which can be adopted both individually and composely (See Fig. 2). The details of each strategy will be illustrated as follows.

**Weighted Summation**. Weighted summation combines the generated structural embedding with the segment in textual and visual embeddings corresponding to the entity in its own feature space, as shown in Fig. 3 (a). The summation procedure can be divided into two parts, *i.e.,* (1) text-structure feature generation and (2) vision-structure feature generation.

More specifically, for the first part, we first extract the specific token representation vector $\mathbf{h}_{e_h}^s \in \mathcal{R}^{1\times D}$ from the structural feature matrix $\mathbf{H}^s \in \mathcal{R}^{N\times D}$ for each entity. Then, the fused feature vector is generated according to Eq. (4).

$$\mathbf{h}_{e_h}^{ts} = \mathbf{h}_{e_h}^t + \lambda_s^{ts} \cdot \mathbf{h}_{e_h}^s, \tag{6}$$

where $\lambda_s^{ts}$ is the weighted hyper-parameter for text-structure feature generation. After that, we replace the feature vector $\mathbf{h}_{e_h}^{ts}$ for $\mathbf{h}_{e_h}^t$ in the original textual feature matrix and generate the text-structure feature matrix $\mathbf{H}^{ts}$.

As for the second part, we first expand structural feature vector $\mathbf{h}_{e_h}^s \in \mathcal{R}^{1\times D}$ to the structural feature matrix $\mathbf{H}_{e_h}^s \in \mathcal{R}^{I\times D}$ based on the Eq. (8) below.

$$\mathbf{H}_{e_h}^s = \mathbf{h}_{e_h}^s \mathbf{1}_I^T, \tag{7}$$

where $\mathbf{H}_{e_h}^s$ has the same dimension as the visual feature matrix $\mathbf{H}_{e_h}^v$ for each entity and $\mathbf{1}_I$ is the row vector with all elements set to 1. This operation aims to expand the vector $\mathbf{h} \in \mathcal{R}^{1\times D}$ to matrix $\mathbf{H} \in \mathcal{R}^{I\times D}$. For example, given $\mathbf{h}$ as [1, 2, 3], we can generate the feature matrix $\mathbf{H}$ as [1, 1, 1; 2, 2, 2; 3, 3, 3].

**Algorithm 1** Psedo-code of SGMPT

**Initialization**: $MKG = (\mathcal{E}, \mathcal{R}, \mathcal{G}, \mathcal{A}_\mathcal{M})$; The text encoder $f_t(\cdot)$; The vision encoder $f_v(\cdot)$; The multi-modal information encoder $f_m(\cdot)$; The structure encoder $g(\cdot)$; Weights in the pre-trained model $W$; Iteration number $t$; Hyper-parameters $\lambda_s^{ts}$, $\lambda_s^{vs}$, $\lambda_a^{ts}$, and $\lambda_a^{vs}$.

1: Generate the structural representation $\mathbf{H}^s$ by Eq. (5);
2: **for** $i = 1$ to $t$ **do**
3:   **for** $e$ in $\mathcal{E}$ **do**
4:     Extract the structural vector $\mathbf{h}_e^s$ for corresponding entity $e$ from $H^s$, and expand it to the matrix $\mathbf{H}_e^s$ by Eq. (7);
5:     Generate the textual representation $\mathbf{H}_e^t$ for target entity by Eq. (3);
6:     Extract the token vector $\mathbf{h}_e^t$ from $\mathbf{H}^t$;
7:     Generate the visual representation $\mathbf{H}_e^v$ for target entity by Eq. (4);
8:     Generate the text-structure vector $\mathbf{h}_{e_h}^{ts}$ by Eq. (6).
9:     Obtain the text-structure feature $\mathbf{H}^{ts}$ by reassigning $\mathbf{h}_{e_h}^{ts}$ to the corresponding position in $\mathbf{H}^t$.
10:     Generate the vision-structure feature $\mathbf{H}_{e_h}^{vs}$ (i.e., $\mathbf{H}^{vs}$) by Eq. (8).
11:     Feed $\mathbf{H}^{ts}$ and $\mathbf{H}_{e_h}^{vs}$ into $f_m(\cdot)$ and get [MASK] embedding $\mathbf{h}_{MASK}$;
12:     Calculate alignment losses $\mathcal{L}_{ts}$ and $\mathcal{L}_{vs}$ by Eq. (9) and Eq. (10);
13:     Calculate cross-entropy loss $\mathcal{L}_{ce}$ referred to [7];
14:     Optimize the whole network $W$ by minimizing $\mathcal{L}$ in Eq. (12);
15:   **end for**
16: **end for**

Then, the fused vision-structure feature matrix $\mathbf{H}^{vs}$ is generated according to Eq. (8).

$$\mathbf{H}_{e_h}^{vs} = \mathbf{H}_{e_h}^v + \lambda_s^{vs} \cdot \mathbf{H}_{e_h}^s, \tag{8}$$

where $\lambda_s^{vs}$ is another weighted hyper-parameter for vision-structure feature matrix generation.

**Alignment Constraint**. Alignment constraint assists in learning unified representations by aligning the textual and visual feature representations to the structural feature representation using a self-supervised alignment loss, *i.e.,* MSE loss [15, 39]. The loss helps pull together different embeddings from different modalities. Similar to the weighted summation, the alignment constraint can also be divided into two parts, *i.e.,* (1) text-structure feature alignment and (2) vision-structure feature alignment.

Concretely, two types of loss functions are adopted, including text-structure alignment loss $\mathcal{L}_{ts}$ and vision-structure alignment loss $\mathcal{L}_{vs}$.

$$\mathcal{L}_{ts} = \mathrm{MSE}(\mathbf{h}_{e_h}^t, \mathbf{h}_{e_h}^s)$$
$$= \left\| \frac{\mathbf{h}_{e_h}^t}{\|\mathbf{h}_{e_h}^t\|_2} - \frac{\mathbf{h}_{e_h}^s}{\|\mathbf{h}_{e_h}^s\|_2} \right\|_2^2 \tag{9}$$
$$= 2 - 2 \cdot \frac{\langle \mathbf{h}_{e_h}^t, \mathbf{h}_{e_h}^s \rangle}{\|\mathbf{h}_{e_h}^t\|_2 \cdot \|\mathbf{h}_{e_h}^s\|_2},$$

$$\mathcal{L}_{vs} = \mathrm{MSE}(\mathbf{H}_{e_h}^v, \mathbf{H}_{e_h}^s)$$
$$= \left\| \frac{\mathbf{H}_{e_h}^v}{\|\mathbf{H}_{e_h}^v\|_2} - \frac{\mathbf{H}_{e_h}^s}{\|\mathbf{H}_{e_h}^s\|_2} \right\|_2^2 \tag{10}$$
$$= 2 - 2 \cdot \frac{\langle \mathbf{H}_{e_h}^v, \mathbf{H}_{e_h}^s \rangle}{\|\mathbf{H}_{e_h}^v\|_2 \cdot \|\mathbf{H}_{e_h}^s\|_2},$$

where $\|\cdot\|$ denotes the 2-norm. Our network is optimized by minimizing the contrastive loss and $\mathbf{H}_{e_h}^s$ is generated by Eq. (8). Moreover, the total loss $\mathcal{L}_a$ for alignment is shown below:

$$\mathcal{L}_a = \lambda_a^{ts} \cdot \mathcal{L}_{ts} + \lambda_a^{vs} \cdot \mathcal{L}_{vs}, \tag{11}$$

where $\lambda_a^{ts}$ and $\lambda_a^{vs}$ are two trade-off hyper-parameters. With the additional alignment losses, the final objective function equals the sum of the cross-entropy loss $\mathcal{L}_{ce}$ with the alignment losses $\mathcal{L}_a$:

$$\mathcal{L} = \mathcal{L}_{ce} + \mathcal{L}_a \tag{12}$$

where the cross-entropy loss is used as the original task loss for link prediction.

## 3.4 Complexity Analysis

In this section, we discuss the computational complexity of our method. According to Algorithm 1, the training process can be divided into several parts: (1) Initialization, (2) Training, and (3) Optimization. We first discuss the complexities of these parts, respectively. We denote the epoch number as $t$, the dimensional size of representations as $D$, the number of entities as $\mathcal{E}$, the number of images for each entity (visual attribute) as $I$, the text tokens number for each entity (textual attribute) as $T$. The total complexity of our model can be formulated as $O(TD + ID + t\mathcal{E}((I^2 + T^2 + IT)D^2 + ITD^2 + I^2D^2 + T^2D^2 + I^2T^2)) = O(t\mathcal{E}((I^2 + T^2 + IT)D^2 + I^2T^2))$. Note that the hyper-parameter $I \leq 10$ is a small constant and can be neglected, and in the actual training, $t \leq 30, T \leq 64, D < 1000$. Thus, the main complexity can be denoted as $O(t\mathcal{E}T^2D^2)$, and this complexity depends mainly on $O(\mathcal{E})$, *i.e.*, the scale of the dataset.

## 3.5 Discussion of SGMPT

We emphasize the attributes of SGMPT to highlight our contributions and novelties. As an attempt at pre-trained transformer models for multi-modal knowledge graph reasoning, we are the first to work to integrate the structural information underlying the MKGs. Inspired by multi-modal information fusion strategies for other tasks, we design two fusion strategies, *i.e.*, weighted summation and alignment constraint, to fuse the extra structural information into the original pre-trained transformer models for MKGR. These two strategies can be used **either independently or composed**. Besides, four trade-off hyper-parameters are introduced to adjust the weights of structural information. Meanwhile, a sensitive analysis is carried out on them. Instead of an independent MKGR model, our SGMPT is more like an **auxiliary mechanism** to assist existing multi-modal pre-trained transformer models for knowledge graph reasoning in achieving better reasoning performance.

## 4 EXPERIMENT

In this section, we first introduce the experimental settings from four aspects, including datasets, evaluation metrics, implementation, and compared baselines. Then, we comprehensively analyze the proposed SGMPT by answering the following questions.

- **Q1: Superiority.** Does SGMPT outperform the existing state-of-the-art existing multi-modal knowledge graph reasoning models, especially for the transformer models?
- **Q2: Effectiveness.** Are the adopted structure encoder and structure-guided fusion modules effective in fusing structure

**Table 2: Statistics of FB15k-237-IMG and WN18-IMG**

| Dataset | #Rel. | #Ent. | #Train | #Dev | #Test |
|---|---|---|---|---|---|
| FB15k-237-IMG | 237 | 14541 | 272115 | 17535 | 20466 |
| WN18-IMG | 18 | 40943 | 141442 | 5000 | 5000 |

information into the MPT model for better MKGR performance?
- **Q3: Efficiency.** Will the additional mechanism for structure information raise the unnecessary model and running time complexity for our SGMPT compared to MKGformer?
- **Q4: Sensitivity.** How does the performance fluctuation of SGMPT with different hyper-parameters?

We conduct experiments to answer the above questions. Specifically, answers of **Q1** to **Q4** are offered in Sec. 4.2 to 4.5.

## 4.1 Experiment Setting

*4.1.1 Datasets.* Two commonly used available datasets, *i.e.,* FB15K-237-IMG and WN18-IMG, for multi-modal knowledge graph relation reasoning are used in this paper. These datasets include three modalities, including text descriptions, corresponding images, and graph structures. Specifically, both FB15k-237-IMG [31] and WN18 [31] datasets are constructed by extending ten images for each entity based on FB15k-237 and WN18. The detailed statistics of these two datasets are shown in Tab. 2.

*4.1.2 Implementation Details.* The experiments are implemented on the computer with an Intel(R) Core(TM) i9-9900K CPU @ 3.60GHz, 64GB RAM, and one GeForce RTX 3090 Ti GPU using PyTorch 1.10.0 in CUDA 11.1. The model parameters are optimized using Adam [22] optimizer, and we conduct a grid search to find suitable hyper-parameters. We select MKGformer as our backbone. Following it, we adopt the BERT base [12] and ViT-B/32 [14] as the text encoder and vision encoder in SGMPT. Besides, a multi-modal information encoder is chosen as the M-Encoder in [7]. As for the structure encoder, most of our experiments are carried out based on the HAKE [53], but we also evaluate the performance of our model with other typical structure encoders, including HousE [27], and COMPGCN [42]. Besides, hyper-parameter $\lambda_s^{ts}$, $\lambda_s^{vs}$, $\lambda_a^{ts}$ and $\lambda_a^{vs}$ are set as 0.01, 0.01, 0.001, and 0.001. Following previous works [7, 43], we use four metrics to evaluate the performance of our model, *i.e.,* (1) Hits@k, where k $\in \{1, 3, 10\}$, and (2) Mean Rank (MR). Besides, the mean results of three runs of each experiment are reported.

*4.1.3 Compared Baselines.* The compared models include two types, *i.e.,* non-transformer KGR models and transformer KGR models. Among the models in the first type, there are five unimodal KGR state-of-the-art models, including TransE [4], DisMult [48], ComplEX [41], ConvE [11], and RGCN [34], and three multi-modal state-of-the-art KGR models, including IKRL [46], TransAE [44] and RSME [43]. As for the transformer models, there are three multi-modal models except the KG-BERT [50], which is the first unimodal KGR model developed based on transformer architecture. Visual-BERT [26] and ViLBERT [30] are the general multi-modal models that can also be applied to multi-modal KGR. Besides, MKGformer is the representative multi-modal KGR model with transformer

**Table 3: Performance comparison of different KGR models for MKGR task on FB15K-237-IMG and WN18-IMG. The best results are in boldface and the second-best results are marked with the underline. Note that the Hit@k is presented in percentage. We reproduce the results of MKGformer since it is the backbone model of our SGMPT.**

| Model | FB15k-237-IMG | | | | WN18-IMG | | | |
|---|---|---|---|---|---|---|---|---|
| | MR | Hits@1 | Hits@3 | Hits@10 | MR | Hits@1 | Hits@3 | Hits@10 |
| *Non-Transformer KGR Models* | | | | | | | | |
| TransE | 323 | 19.8 | 37.6 | 44.1 | 357 | 4.0 | 74.5 | 92.3 |
| DisMult | 512 | 19.9 | 30.1 | 44.6 | 665 | 33.5 | 87.6 | 94.0 |
| ComplEx | 546 | 19.4 | 29.7 | 45.0 | - | 93.6 | 94.5 | 94.7 |
| ConvE | 249 | 22.5 | 34.1 | 49.7 | - | 41.9 | 47.0 | 53.1 |
| RGCN | 600 | 10.0 | 18.1 | 30.0 | - | 8.0 | 13.7 | 20.7 |
| IKRL(UNION) | 298 | 19.4 | 28.4 | 45.8 | 596 | 12.7 | 79.6 | 92.8 |
| TransAE | 431 | 19.9 | 31.7 | 46.3 | 352 | 32.3 | 83.5 | 93.4 |
| RSME(ViT-B/32+Forget) | 417 | 24.2 | 34.4 | 46.7 | - | **94.3** | 95.1 | - |
| *Transformer KGR Models* | | | | | | | | |
| KG-BERT | **153** | - | - | 42.0 | 58 | 11.7 | 68.9 | 92.6 |
| VisualBERT | 592 | 21.7 | 32.4 | 43.9 | 122 | 17.9 | 43.7 | 65.4 |
| ViLBERT | 483 | 23.3 | 33.5 | 45.7 | 131 | 22.3 | 55.2 | 76.1 |
| MKGformer | 252 | 24.3 | 36.0 | 49.9 | **25** | 93.5 | 95.8 | 97.0 |
| SGMPT (Ours) | 238 | **25.2** | **37.0** | **51.0** | 29 | **94.3** | **96.6** | **97.8** |

**Table 4: Ablation study of SGMPT on FB15K-237-IMG. 'WS' and 'AC' represent the weighted summation and alignment constraint. The Hit@k is presented in percentage.**

| Model | FB15k-237-IMG | | | |
|---|---|---|---|---|
| | MR | Hits@1 | Hits@3 | Hits@10 |
| SGMPT | 238 | 25.2 | 37.0 | 51.0 |
| - $WS^{ts}$ | 242 | 24.7 | 36.6 | 50.6 |
| - $WS^{vs}$ | 240 | 25.0 | 36.8 | 50.8 |
| - $AC^{ts}$ | 242 | 24.7 | 36.6 | 50.6 |
| - $AC^{vs}$ | 241 | 24.8 | 36.8 | 50.7 |
| - $(WS^{ts}$ & $WS^{vs})$ | 247 | 24.5 | 36.3 | 50.4 |
| - $(AC^{ts}$ & $AC^{vs})$ | 248 | 24.4 | 36.2 | 50.2 |
| - $(WS^{ts}$ & $AC^{ts})$ | 248 | 24.4 | 36.2 | 50.2 |
| - $(WS^{vs}$ & $AC^{vs})$ | 245 | 24.6 | 36.5 | 50.5 |
| - $(WS^{ts}$ & $WS^{vs}$ & $AC^{ts}$ & $AC^{vs})$ | 252 | 24.3 | 36.0 | 49.9 |

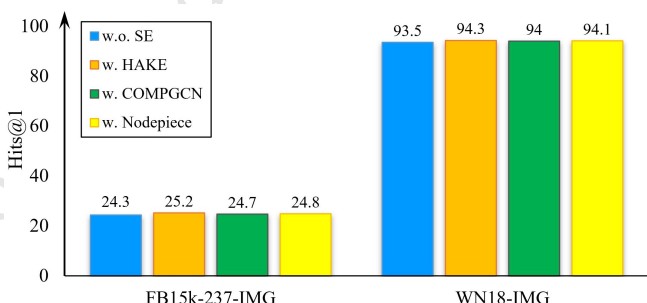

**Figure 4: Performance comparison with different structure encoders. 'w.' and 'w.o.' represent the with and without separately. 'SE' represents the structure encoder.**

architecture. However, none of the transformer KGR models makes use of the structure information except for our SGMPT.

## 4.2 Performacne Comparsion (RQ1)

The overall performance comparison is carried out to answer **Q1**. We compare our SGMPT with thirteen other state-of-the-art models on two benchmark datasets. The results in Tab.3 show that our SGMPT outperforms all of the evaluation metrics compared to non-transformer KGR models. Besides, compared to the transformer KGR models, our SGMPT outperforms most of the evaluation metrics compared to transformer KGR models. In particular, it is more apparent that our model boosts the performance on FB15-237-IMG. Concretely, our model makes 4.1%, 7.6%, and 2.6% improvements on Hits@1, Hits@3, and Hits@10 compared to non-transformer KGR models, and 3.7%, 2.8%, and 2.2% improvements on Hit@1, Hits@3, and Hits@10 compared to transformer KGR models, respectively. It shows the superiority of our model SGMPT. Besides, it also indicates the advances of the transformer paradigm for the MKGR task, since the average performances of transformer KGR models are better than non-transformer KGR models in the multi-modal scenario. Although the MR values of our model are not the best, they still occupy the second-best positions. Moreover, our MR performances are actually better and comparable compared to the backbone MPT model, *i.e.*, MKGformer. The above results show

that our SGMPT can achieve better reasoning performance in the multi-modal scenario, which further indicates the effectiveness of involving the structural information in MPT models.

## 4.3 Ablation Study (RQ2)

The ablation studies are conducted on FB15-237-IMG to answer the Q2. Specifically, two subquestions need to be answered in the following subsections, *i.e.,* **(1)** "Can the structure-guided fusion modules make differences?" and **(2)** "Will different structure encoders benefit the structure information fusion procedure?".

*4.3.1 Effectiveness of the Structure-guided Fusion Module.* We evaluate the effectiveness of the structure-guided fusion module from four parts, *i.e.,* weighted summation, alignment constraint, text-structure fusion, and vision-structure fusion. Concretely, ten sub-models are compared, including (1) the original SGMPT model, (2) SGMPT without text-structure weight summation denoted as "- $WS_{ts}$", (3) SGMPT without vision-structure weight summation denoted as "- $WS_{vs}$", (4) SGMPT without text-structure alignment constraint denoted as "- $AC_{vs}$", (5) SGMPT without vision-structure alignment constraint denoted as "- $AC_{vs}$", (6) SGMPT without weight summation denoted as "- $(WS^{ts}$ & $WS^{vs})$", (7) SGMPT without alignment constraint summation denoted as "- $(AC^{ts}$ & $AC^{vs})$",

**Table 5: The comparison of model parameters and running time between our SGMPT and backbone MKGformer. Note that H represents hour.**

| Model | FB15k-237-IMG | | WN18-IMG | |
|---|---|---|---|---|
| | # Param. | Time | # Param. | Time |
| MKGformer | 950.540 M | 9.6 H | 1029.818 M | 24.3 H |
| SGMPT | 956.687 M | 9.8 H | 1032.893 M | 25.4 H |

(8) SGMPT without text-structure strategies denoted as "- (WS$^{ts}$ & AC$^{ts}$)", (9) SGMPT without vision-structure strategies denoted as "- (WS$^{vs}$ & AC$^{vs}$)", (10) SGMPT without structure-guided fusion module denoted as "- (WS$^{ts}$ & WS$^{vs}$ & AC$^{ts}$ & AC$^{vs}$)". Tab. 5 shows that performance boosts are made by the strategies used in the structure-guided fusion module, *i.e.,*, 5.9%, 3.7%, 2.8%, and 2.2% improvements for MR, Hits@1, Hits@3, and Hits@10. More specifically, the alignment constraint and weighted summation are relatively equivalently effective to MKGR. Besides, vision-structure fusion is less than text-structure fusion, due to the information loss caused by the vector expanding operations. In all, the promising results demonstrate the effectiveness of the module and also prove that strategies can be adopted either individually or composedly.

*4.3.2 Influence of Different Structure Encoder.* We also replace the structure encoder $g(\cdot)$, *i.e.,* HAKE [53], as three other typical structure encoders, including CompGCN [42], and Nodepiece [17]. Experiments are further conducted based on FB15k-237-IMG for Hit@1 and Hit@10 (See Fig. 4). It indicates that various structure encoders can all benefit the structure information fusion procedure, *i.e.,* on average 2.5% and 2.0% improvements on FB15k-237-IMG and WN18-IMG separately. Besides, the HAKE is the most effective choice among these three structure encoders.

Based on promising results and the above analyses in Sec 4.3.1 and Sec. 4.3.2, we can assert that both the adopted structure encoder and designed structure-guided fusion modules are effective in fusing the omitted structure information in KG into the MPT model for better MKGR performance.

## 4.4 Efficiency Analysis (RQ3)

We analyze the complexity of our SGMPT from two aspects, *i.e.,* parameter number and the running time of the models. Concretely, the comparison is carried out between our SGMPT and the backbone MKGformer on both FB15k-237-IMG and WN18-IMG. According to Tab. 4, it is observed that the efficiency of our model is a little bit worse than MKGformer, *i.e.,* average 0.45% and 3.3% increasing on the number of parameters and running time, respectively. Since the structure fusion mechanism is designed to complement the omitted structure information, our SGMPT will definitely be more complex and time-consuming. However, considering the performance improvements for MKGR, it is acceptable with the comparable complexity, which indicates that our SGMPT does not raise unnecessary parameters and running time.

## 4.5 Sensitivity Analysis (RQ4)

We investigate the influence of the hyper-parameter $\lambda_s^{ts}$, $\lambda_s^{vs}$, $\lambda_a^{ts}$, $\lambda_a^{vs}$ on FB15k-237-IMG for Hit@1 and Hit@10. Besides, the scope of these four hyper-parameters is selected in $\{0.001, 0.01, 0.1, 1\}$.

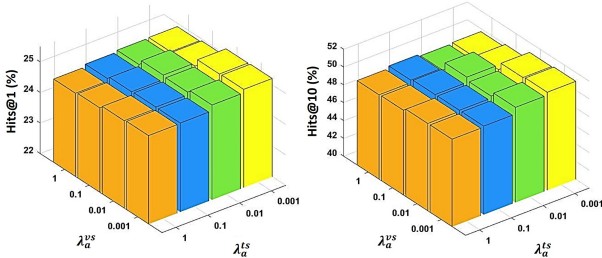

(a) hyper-parameters analysis for weighted summation.

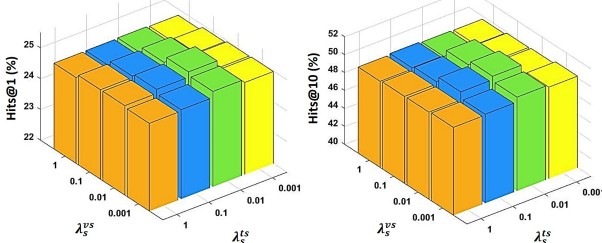

(b) hyper-parameters analysis for alignment constraint.

**Figure 5: Sensitivity analysis of hyper-parameter $\lambda_s^{ts}$, $\lambda_s^{vs}$, $\lambda_a^{ts}$, $\lambda_a^{vs}$ on FB15k-237-IMG for Hit@1 and Hit@10. Note that $\lambda_a^{ts}$, $\lambda_a^{vs}$ both equal to 0.001 in subgraph (a) and $\lambda_s^{ts}$, $\lambda_s^{vs}$ both equal to 0.01 in subgraph (b).**

We observe that the MKGR performance will not fluctuate greatly when the parameter is varying in Fig. 5. It demonstrates that the performance of SGMPT is insensitive to these hyper-parameters. We can further find out that best performances are reached with the combination of $\lambda_s^{ts}$, $\lambda_s^{vs}$, $\lambda_a^{ts}$ and $\lambda_a^{vs}$ set to (0.01, 0.01, 0.001, 0.001).

## 5 CONCLUSION

In this paper, we propose a novel and simple graph Structure Guided Multi-modal Pre-trained Transformer model for knowledge graph reasoning, termed SGMPT. As the first multi-modal pre-trained transformer model to leverage the structure information for multi-modal knowledge graph reasoning, SGMPT is the first work to add a specific structure information fusion procedure based on the MPT backbone. The structure encoder and structure-guided fusion module are required to complete the procedure. More concretely, various KGE models can be selected as the structure encoder, and two different strategies, *i.e.,* weighted summation and alignment constraint, can be adopted both individually and composedly. Extensive experiments on two benchmark datasets, *i.e.,* FB15k-IMG and WN18-IMG, demonstrate the promising capacity of our SGMPT from four aspects, including superiority, effectiveness, complexity, and sensitivity.

In the future, we plan to continue to improve the capacity of our model toward its limitations. For example, four hyperparameters are enrolled to complement the omitted structural information. Moreover, the weighted summation strategy can be more fine-grained, though the current simple addition can also achieve promising improvements.

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
