# OpenReview forum: "Simple Yet Effective: Structure Guided Pre-trained Transformer for Multi-modal Knowledge Graph Reasoning"
_acmmm.org/ACMMM/2024/Conference — MM2024 Poster_

### Official Review · Reviewer_rg6E · 2024-05-24

**Rating:** 5
**Confidence:** 3

**Summary:**

A graph structure-guided multimodal pre-training framework using the Transformer model for knowledge graph reasoning is proposed to supplement the omitted graph structure information used for knowledge graph reasoning with a structure encoder and a structure-guided fusion module.

**Strengths:**

1. By introducing a graph structure encoder and a structure-guided fusion module, the structural information in the knowledge graph is utilized for the first time to improve the inference ability of the model.

2. The design is simple but effective, fusing structural information into textual and visual features through two strategies, weighted summation and alignment constraints, to improve the inference performance of the model.

**Limitations:**

1. The performance improvement of the experimental results is not very high, not very convincing, and even less than 1% on the WN18-IMG dataset.

2. Whether the baseline for comparison of experimental results is available for the last two years of the method.

**Suitability:**

3

---

### Official Review · Reviewer_rLCt · 2024-05-24

**Rating:** 5
**Confidence:** 3

**Summary:**

This paper introduces a novel method named SGMPT (graph Structure Guided Multi-modal Pre-trained Transformer) with a graph structure encoder and structure-guided fusion module. The method is simple yet effective, which outperforms existing state-of-the-art methods on two multimodal knowledge graph reasoning datasets FB15k-237-IMG and WN18-IMG.

**Strengths:**

1. The idea that mines structure information for multimodal knowledge graph reasoning is reasonable and effective intuitively.
2. The proposed structure-guided fusion module fuses structural information with modal information in a very simple and intuitive way.
3. The results on two multimodal knowledge graph reasoning datasets FB15k-237-IMG and WN18-IMG demonstrate the effectiveness of the method.
4. The ablation study and analysis are comprehensive and specific, and the experimental organization is reasonable.

**Limitations:**

1. The text encoder and the visual encoder are relatively simple. It would be interesting to experiment with larger models, such as LLaVA.
2. The initialization of the multi-modal information encoder is important. What is used to initialize it?

**Suitability:**

3

---

### Official Review · Reviewer_PzH7 · 2024-05-24

**Rating:** 3
**Confidence:** 3

**Summary:**

This paper introduces a new model called Structure Guided Multi-modal Pre-trained Transformer (SGMPT) for knowledge graph reasoning. It addresses the limitations of previous multi-modal pre-trained transformers (MKGformer) by fully utilizing the rich structural information present in multi-modal knowledge graphs .
SGMPT incorporates a graph structure encoder to encode structural features. It also includes a structure-guided fusion module that uses weighted summation and alignment constraint strategies to inject the structural information into both textual and visual features.

**Strengths:**

SGMPT addresses the limitation of previous multi-modal pre-trained transformers by fully utilizing the rich structural information present in multi-modal knowledge graphs.
It incorporates a graph structure encoder and a structure-guided fusion module to inject the structural information into both textual and visual features.

**Limitations:**

1. The contribution of this paper is limited, as it is an incremental work based on MKGformer. The basic framework of this paper follows MKGformer completely, with the addition of a graph-guide branch.

2. One curious point is that while MKGformer performs multiple tasks (multimodal link prediction, multimodal RE, and multimodal NER), why this paper only focuses on link prediction?

3. Where does the viewpoint come from that "Among the MPT models for KGR, MKGformer is the most representative model with the best reasoning capacity"? Its performance is not better than some of the latest methods, so how is the "best reasoning capacity" evaluated?

4. Why is there such a significant difference between the reported experimental results of MKGformer in the FB15k-237-IMG dataset (Hits@10 = 0.573) and the results in this paper (Hits@10 = 0.499)?

5. The paper lacks some necessary citations and comparison experiments. It fails to compare the proposed method with the latest state-of-the-art methods. It would have been beneficial to include comparisons with recent approaches such as QuatE [1] for uni-modal KGC, CSProm-KG [2] and SimKGC [3] for description-based KGC, and LAFA [4], MMNER [5], AdaMF-MAT [6] for multimodal KGC.

6. The three-dimensional histogram in Figure 5 makes it difficult to extract information and judge the effectiveness of different parameters. It is recommended to replace it with a heatmap.




[1] Quaternion Knowledge Graph Embeddings

[2] Dipping PLMs Sauce: Bridging Structure and Text for Effective Knowledge Graph Completion via Conditional Soft Prompting

 [3] Simkgc: Simple contrastive knowledge graph completion with pre-trained language models

 [4] LAFA: Multimodal Knowledge Graph Completion with Link Aware Fusion and Aggregation

[5] Relation-enhanced negative sampling for multimodal knowledge graph completion

 [6] Unleashing the Power of Imbalanced Modality Information for Multi-modal Knowledge Graph Completion

**Suitability:**

3

---

### Official Review · Reviewer_BAnG · 2024-05-29

**Rating:** 5
**Confidence:** 2

**Summary:**

This paper introduces a structure guided multimodal pre-trained Transformer model called SGMPT for knowledge graph inference tasks. Unlike previous knowledge graph inference models, this model can utilize the structural information in the knowledge graph and inject structural information into text and visual features by introducing structural encoding and structural guidance fusion modules. The experimental results show that the model performs better than the existing best models on the FB15k-237-IMG and WN18-IMG datasets, and demonstrate the effectiveness of the design strategy.

**Strengths:**

Strengths:
1. Traditional multi-modal pre-training models only consider the relationship between textual and visual information, overlooking the significance of structural information. The SGPMT model, however, introduces structural information and employs a structure-guided fusion module, enabling the model to better leverage multimodal information and enhance its performance. SGMPT stands as the first multi-modal pre-training Transformer designed for Knowledge Graph Reasoning (KGR).

2. SGMPT designs two effective structure-guided fusion strategies: weighted summation and alignment constraint, which can be adjusted according to the demands of different tasks, thereby further enhancing the model's adaptability and generalization capability.

3. Acting as a plug-and-play fusion mechanism, SGMPT allows for the flexible selection of various Knowledge Graph Embedding (KGE) models as structural encoders.

4. Experimental results demonstrate that SGMPT achieves superior performance across two public datasets, affirming the efficacy and superiority of the SGMPT model.

**Limitations:**

1.The representation in Figure 5 is not very clear and could be replaced with a more effective visualization method.

2.Although a simple additive approach currently yields noticeable improvements, the weighted summation strategy appears rather rudimentary.

3.Ideally, illustrating the attention weights assigned to the text-structure features and vision-structure features after their fusion with structure information would provide a clearer understanding of precisely how the incorporated structural information strengthens the connections between entities.

**Suitability:**

3

---

### Meta-Review · Area_Chair_JvNG · 2024-07-04

**Recommendation:** Accept (Poster)
**Confidence:** 5

**Metareview:**

In this paper, the authors have proposed a novel and simple graph Structure Guided Multi-modal Pre-trained Transformer model for knowledge graph reasoning, termed SGMPT. All the reviewers recognize the contributions/novelties of this paper. However, they have pointed out some limitations. The authors are expected to address these limitations in the camera ready version.